# New Insights in the Control of Fat Homeostasis: The Role of Neurotensin

**DOI:** 10.3390/ijms23042209

**Published:** 2022-02-17

**Authors:** Ilaria Barchetta, Marco Giorgio Baroni, Olle Melander, Maria Gisella Cavallo

**Affiliations:** 1Department of Experimental Medicine, Sapienza University, 00161 Rome, Italy; ilaria.barchetta@uniroma1.it; 2Department of Clinical Medicine, Public Health, Life and Environmental Sciences, University of L’Aquila, 67100 L’Aquila, Italy; marcogiorgio.baroni@univaq.it; 3Neuroendocrinology and Metabolic Diseases, IRCCS Neuromed, 86077 Pozzilli, Italy; 4Department of Clinical Sciences Malmö, Lund University, 20213 Malmö, Sweden; olle.melander@med.lu.se; 5Department of Emergency and Internal Medicine, Skåne University Hospital, 20213 Malmö, Sweden

**Keywords:** neurotensin, gut peptides, gastrointestinal hormones, obesity, NAFLD, fatty liver, insulin resistance, type 2 diabetes

## Abstract

Neurotensin (NT) is a small peptide with pleiotropic functions, exerting its primary actions by controlling food intake and energy balance. The first evidence of an involvement of NT in metabolism came from studies on the central nervous system and brain circuits, where NT acts as a neurotransmitter, producing different effects in relation to the specific region involved. Moreover, newer interesting chapters on peripheral NT and metabolism have emerged since the first studies on the NT-mediated regulation of gut lipid absorption and fat homeostasis. Intriguingly, NT enhances fat absorption from the gut lumen in the presence of food with a high fat content, and this action may explain the strong association between high circulating levels of pro-NT, the NT stable precursor, and the increased incidence of metabolic disorders, cardiovascular diseases, and cancer observed in large population studies. This review aims to provide a synthetic overview of the main regulatory effects of NT on several biological pathways, particularly those involving energy balance, and will focus on new evidence on the role of NT in controlling fat homeostasis, thus influencing the risk of unfavorable cardio–metabolic outcomes and overall mortality in humans.

## 1. Introduction

Neurotensin (NT) is a biologically active, 13 amino acid peptide derived from the intracellular proteolytic cleavage of its precursor in two fractions; the longer one is constituted of 163 amino acids and is known as proneurotensin (pro-NT) [1]. Intracellular proteolytic processing is mediated by members of the prohormone convertase family [2]. The active NT peptide is then stored in dense-core vesicles and released in a calcium-dependent manner [3,4].

NT and pro-NT are released in equimolar amounts and share the same biological activity; since pro-NT is characterized by greater stability than NT in the bloodstream, the measurement of this longer fragment is considered as the most reliable method for assessing circulating NT concentration [1].

NT exerts its biological actions by the linkage with three specific receptors named neurotensin receptor-1 (NTSR1), -2 (NTSR2), and -3 (NTSR3), the latter also being known as sortilin-1 [5]. As a consequence, NT effects largely depend on the receptor type and distribution within tissues and organs.

While NTSR1 and NTSR2 are classical G-protein–coupled receptors (GPCRs), NTSR3/sortilin is a single-pass transmembrane neuropeptide which acts by sorting intracellular proteins [6]. NTSR1 is recognized as the main receptor that, among other central functions, influences energy balance [7] being expressed at both the central and peripheral level, mainly in the gastrointestinal tract [8]. NTSR2 is found in many brain areas and the upper intestinal tract, whereas NTSR3 is predominantly expressed in the periphery [8].

NT was initially identified in the hypothalamus and described as a neuropeptide transmitter [9]. In relation to the large number of brain circuits and hormone responses that NT mediates, this neuropeptide is involved in the central regulation of a broad range of biological functions, from appetite and activity behavior, to thermogenesis, nociception, blood pressure, reward mechanisms, sleep, and several others [10,11].

NT is also released from intestinal neuroendocrine cells in response to dietary fat content and increased lipid concentrations in the gut lumen [12,13]. Once secreted, NT facilitates lipid absorption and regulates the secretion of other gastrointestinal peptides via endocrine and paracrine circuits [12,13,14,15,16].

Although NT is primarily produced in the hypothalamus, in the pituitary, and in neuroendocrine cells of the ileum, it was also demonstrated to be expressed in nerve fibers in the heart and adrenal medulla [17,18]. Furthermore, the NT gene is transiently expressed in fetal tissues of organs like the liver, pancreas, and colon [19]; NT expression was also shown in tumoral cells of the colon, lung, pleura, pancreas, breast, and in melanoma cells [20].

Peripheral NT is involved in the regulation of immune–inflammatory processes; it stimulates cell proliferation and contributes to the development of site-specific and systemic pro-inflammatory conditions, which in turn may negatively impact on metabolic processes and promote dysmetabolic diseases and carcinogenesis [21,22,23,24,25].

Lastly, evidence showed that peripheral NT is able to cross the blood–brain barrier [26,27,28]. Once penetrated in the central nervous system, NT may have an influence on other hypothalamic and pituitary pathways controlling energy balance and body weight, i.e., by inhibiting the thyrotropin-releasing hormone (TRH) [29] and stimulating the secretion of the corticotropin-releasing hormone (CRH) [30].

Indeed, the overall control exerted by NT on the major metabolic pathways is integrated and accounts for central and peripheral actions, exerted through direct and indirect mechanisms.

## 2. Central NT in the Regulation of Metabolism

Beside the regulation of primary homeostatic functions, i.e., thermogenesis, pain, sleep control, as well as blood pressure support, a role for NT in controlling food intake and energy balance has recently emerged [31,32,33].

The central effects exerted by NT on metabolism are explained by its role as a neuropeptide transmitter in dopaminergic areas. Experimental data showed that in animal models, the pharmacological inoculation of NT in brain areas within the limbic system induced changes in food intake and activity behavior, and the effects differed in relation to the specific area involved [33,34,35,36].

In general, NT in the brain has a pro-anorexigenic effect. As proof of this, experimentally induced obesity was found to be associated with reduced brain NT signaling; diminished NT expression explained overeating and weight gain in obese rodents [37,38,39,40].

However, whole body knockout mice for the NT gene are protected from obesity, as a result of the loss of peripheral NT activity on gut lipid absorption; no change in food intake and physical activity were observed in these models [41].

Indeed, all data available thus far suggest that the contribution of central NT may not be sufficient for determining overall metabolic phenotype, although NT modulation could modify it by influencing appetite and/or locomotor activity.

The involvement of brain NT in the regulation of energy balance, whose detailed description is beyond the purpose of this review, has been recently reviewed by Ramirez and Lenninger [33]. The main effects of central NT on metabolism are summarized in Table 1.

**Table 1 ijms-23-02209-t001:** Experimental NT injection and major central metabolic responses. A green color indicates favorable effects on metabolism, while effects that negatively impact on metabolism are marked with an orange background. Data are from rat models.

Brain Area Involved	NT Effects	References
Ventral tegmental area	Reduced feeding Increased activity	Hawkins et al. [34] Kelley et al. [35] Kalivas et al. [42]
Substantia nigra	Reduced feeding	Vaughn et al. [36]
Nucleus accumbens	Reduced activity Increased resting behavior No effects on feeding	Kalivas et al. [43] Ervin et al. [44] Robledo et al. [45]
Hippocampus	Increased activity	Cador et al. [46]

## 3. NT as a Gut Hormone—Lipids and Fat Homeostasis

### 3.1. NT and Lipid Metabolism

The identification of NT as a gut peptide and its localization in human enteroendocrine (N) cells came from studies performed between the 1970s and 1980s [12,13,14,15,16].

The concentration of this short gut peptide was shown to increase from duodenum to distal ileum [12,13].

Only a few years later, the primary function of intestinal NT was described as promoting intestinal fat absorption [14,15,16]. In healthy volunteers, jejunal infusion of short-, medium-, and long-chain fatty acids resulted in an increase in blood NT concentration, with a peak after 60 min from infusion [14,15].

Conversely, NT administration was shown to delay gastric emptying and to induce gastric acid and pepsin output [16].

Several studies confirmed the strong positive correlation between fat ingestion and blood NT increases [47,48,49]. Lastly, Fawad et al. demonstrated that the amount of rise of plasma NT after fat ingestion parallels blood triglycerides upsurge [49], pointing to a potential contribution of NT to intestinal lipid uptake. Plasma pro-NT peak was reached two to three hours after saturated or unsaturated fat ingestion, respectively, and a strong direct correlation between blood pro-NT and triglycerides was observed, regardless of the form of lipid load [49].

Fat ingestion also stimulates the activation of the NT/NTSRs pathway in the long term, as demonstrated in rats, where prolonged high fat diet (HFD) feeding disrupts peripheral NT signaling [21]. In this investigation, chronically elevated fat ingestion increased basal plasma pro-NT levels and the gut expression of NTSR1 in comparison to what was observed in rats fed with a control diet. In the long term, brain NTSR1 expression also increased, whereas no change was observed in brain NT levels [21].

Indeed, as an overall result, fat-rich diet ingestion increases pro-NT, which, in turn, enhances the activity of gut NTSR1, resulting in augmented fat absorption, visceral fat, and body weight [21,41].

Several investigations explored the mechanisms behind NT secretion from intestinal enteroendocrine cells in relation to high gut fat content [50,51,52,53].

NT secretion is promptly stimulated by high intestinal free fatty acid concentrations, particularly by long-chain fatty acids (LCFAs), such as docosahexaenoic acid (DHA) [48], and is mediated by the free fatty acid receptors-1 (FFAR1, previously named GPR40) and -4 (FFAR4, previously named GPR120). These are G-protein-coupled receptors that are widely expressed in enteroendocrine cells and regulate the secretion of gut hormones [54] and insulin [55]. Both receptors mediate the absorption of saturated and unsaturated medium-chain fatty acids (MCFAs) and LCFAs. FFAR1 directly promotes insulin secretion: once activated by MCFA and LCFA, FFAR1s expressed in pancreatic *β* cells activate the G-protein, which, in turn, transduces the signal leading to the stimulation of insulin secretion through the elevation of intracellular calcium [56]. Conversely, FFAR4 has a more critical role in NT release [50], as well as in the secretion of glucagon-like peptide 1 (GLP1), which in turn indirectly influences insulin secretion as well [57].

FFAR4s are also expressed by macrophages and adipocytes, which positively modulate immune response and insulin sensitivity [58].

The regulation of NT secretion involves important crosstalk converging on the extracellular signal-regulated kinases (ERK)/mitogen-activated protein kinases (MAPK), 5′ adenosine monophosphate-activated protein kinase (AMPK), and the mammalian target of rapamycin complex (mTORC) signaling pathways, which are central regulators of molecular hormonal signaling [51,52,53,57]. FFAR’s activation by MCFAs and LCFAs induces a rise in intracellular Ca^2+^ concentration and then the activation of MAPK/ERK1/2, which therefore represent the effectors of intracellular signaling mediated by FFAR4 and FFAR1, leading to NT secretion [57].

Conversely, AMPK was shown to modulate NT secretion after lipid load in a more intricate way [51,52,53,54]. While some data point towards a positive regulation of NT release by AMPK activation, exerted by the inhibition of mTORC1 signaling [51,52,53], other evidence shows that AMPK acts as a negative regulator in the pathway, and that FFAR1 and FFAR4 activation by fatty acids drives NT release in the gut lumen [54]. In this study, treatment with an AMPK inhibitor enhanced NT release by increasing *p-*ERK1/2 activation [54].

Therefore, data thus far show the existence of inhibitory crosstalk in the regulation of NT secretion between MAPK and AMPK signaling pathways downstream of FFAR1 and FFAR4. MAPK plays a dominant role, whereas AMPK negatively regulates NT secretion, mediated by FFAR1/FFAR4-MAPK activation [54].

Indeed, the downstream signaling cascade which finalizes NT secretion from neuroendocrine intestinal cells reflects the pivotal role of this peptide in metabolic regulation in the presence of a high amount of energy substrate, such as after lipid load.

AMPK signaling also takes part in NT-mediated lipid absorption. Once secreted, NT augments fatty acid uptake by intestinal mucosal cells through the NTSR1/NTSRT3-mediated inhibition of AMPK [41], which is known to restore energy imbalance and to attenuate lipogenesis [59].

However, not only direct mechanisms contribute to the NT-mediated promotion of intestinal lipid absorption [60,61,62,63,64]. Several investigations have described how NT orchestrates the cascade of processes from fat ingestion that leads to augmented fatty acid uptake and rise in blood lipids. Those mechanisms mainly involve bile acid output and enterohepatic circulation [60], although some effects are also exerted by the NT-mediated reduction of small bowel motility [61], modulation of local blood flow [62], and enhanced pancreatic exocrine output [63,64].

These effects may also lead to pathophysiological drift: chronically high gut NT levels are associated with the development of local pro-inflammatory milieu and unfavorable microbiota composition, with a negative impact on intestinal mucosa integrity and associated increased permeability (see also Section 3.3).

### 3.2. NT and Bile Acid Metabolism

Among the determinants of enhanced lipid absorption after a rise in NT concentration, the modulation of bile acid release and uptake plays a primary role. Twenty years ago, Gui et al. [65] produced the first evidence that NT enhances intestinal absorption of taurocholic acid, the most represented conjugated bile acid in rats, without altering the rate of uptake of cholic acid, its unconjugated counterpart. Other experiments also demonstrated a stimulatory effect of NT on gallbladder motility [66], promoting the flow of bile from the biliary tract to the duodenum [67].

Since intestinal absorption is the rate-limiting step of bile acid enterohepatic circulation, NT-induced greater bile acid availability represents a mechanism to promote lipid emulsification and micelle formation, finally resulting in enhanced fat absorption from the gut lumen.

Bile acid circulation is altered in obesity [68,69,70]. Recent experiments investigated the role of NT in mediating the association between HFD-induced decreases in plasma bile acid levels and the presence of obesity, using whole body knockout mice for the NT gene [71]. These studies demonstrated that primary and secondary bile acids were downregulated by HFD in wild-type mice, whereas NT deficiency improved the bile acid pool, attenuating this defect [71].

NT is also directly involved in the regulation of the farnesoid X receptor (FXR), a major regulator of bile acid homeostasis and enterohepatic circulation [72].

In the ileum, FXR regulates the uptake of bile acids by controlling the expression of their specific transporters [73]. FXR overexpression and mimic peptides ameliorate glucose tolerance, insulin sensitivity, and lipid profile in experimental models [74,75]. NT was shown to directly stimulate FXR and bile acid transporter signaling under non-obese conditions, whereas it exerts the opposite effects in obesity, contributing to the alteration of the metabolic phenotype associated with weight excess [71]. Conversely, no data are available on the regulation of hepatic bile acid synthesis, which in turn is mediated by liver X receptor (LXR) signaling, via the NT/NTSRs axis. Similarly, despite its role in favoring bile acids’ enterohepatic circulation, no clear evidence is available in humans regarding a potential direct influence of NT on cholesterol metabolism.

### 3.3. NT, Microbiota Composition, and Gut Mucosal Homeostasis

As for gut lipid metabolism, it is worth mentioning some emerging evidence on the contribution of NT in the development of gut mucosal alternations and increased permeability in HFD-induced obesity [76].

Mice knocked-out for the NT gene were protected from HFD-induced gut dysbiosis compared to wild-type mice regarding microbial diversity and the *Firmucudes/Bacteriodes* ratio, suggesting an additional mechanism linking NT to dysmetabolic conditions [76,77].

Fat-rich diets are known to alter microbiota composition and mucosal integrity, disrupting the Mmp7/α-defensin axis and increasing local inflammation. Of note, obese mice whole body knocked-out for the NT gene are protected from the reduction of α-defensin expression and dysbiosis caused by HFD feeding [76].

Indeed, NT was demonstrated to downregulate the expression of the α-defensin gene, and is also involved in NF-κB signaling via the modulation of protein kinase C (PKC) [76].

Therefore, among other functions, initial evidence shows a role of NT in enhancing the alterations in gut homeostasis that occur in the presence of chronically high fat ingestion and obesity. These processes include gut dysbiosis and loss of membrane integrity; however, the global evidence linking NT to microbiota composition is limited. Other data point to a direct role of NT in controlling gut permeability, via a mechanism involving the regulation of defensins [76] in obesity.

A schematic representation of the mechanisms mediated by gut NT that promote intestinal lipid absorption is provided in Figure 1:

### 3.4. NT and Adipose Tissue

The involvement of NT in fat metabolism is not only relative to the promotion of gut fat absorption and the rise of blood lipids after lipid load. Some studies show that NT also takes part in lipid trafficking, inflammatory responses, and browning in adipose tissue, thus influencing its overall metabolism.

Elevated NT/NTSR levels promote the accumulation of visceral fat, adipocyte hypertrophy, and adipose tissue inflammation, as demonstrated in animal models of NT knockout and overexpression [41]. In this study, the loss of the NT gene in HFD-fed mice prevented excessive body weight gain and adipocyte enlargement, and decreased inflammatory infiltrates and numbers of macrophages, all which are phenomena that are associated with metabolic dysfunction [78,79,80]. Indeed, the capability of NT knockout to preserve the structural integrity of adipose tissue under hypercaloric and fat-rich diet translates into better glucose tolerance and insulin sensitivity [41].

A relationship between high pro-NT and adipose tissue inflammation was also observed cross-sectionally in the clinical setting. Indeed, in a clinical investigation conducted in obese individuals, higher pro-NT levels were associated with histological features and gene expression patterns of inflammation, such as greater macrophage infiltration and hypoxia, and insulin resistance in omental white fat. This resulted in a worse metabolic profile, a higher prevalence of type 2 diabetes (T2DM) and/or non-alcoholic fatty liver disease (NAFLD) in individuals with elevated pro-NT, regardless of body weight per se [81].

In addition to its role in enhancing fat absorption, favoring lipid accumulation, and adipose tissue expansion/inflammation, some studies have explored whether NT also exerts direct effects on adipocytes and adipose tissue.

Indeed, NT modulates the blood flow in human adipose tissue [82]. Intravenous infusion of labeled NT at a concentration in the range obtained during fatty meals reduced the blood flow in abdominal fat but not in the thigh. Moreover, the vasoconstriction observed after NT infusion negatively correlated with the index of adiposity. Overall, this was interpreted by the authors as a potential mechanism exerted by NT for regulating the post-prandial uptake of substrates—i.e., lipids—from adipose tissue in the post-prandial phase [82].

Increased NT and NTSR1 expression have been detected in mesenteric and peri-intestinal adipose tissue in the condition of colitis, and treatment with NT was able to induce IL-6 secretion from preadipocytes [22].

Very recently, the expression of NT and NTRS2 genes has been detected in lymphatic endothelial cells within mesenteric adipose tissue [71]. Lymphatic endothelial cells from mice and humans were demonstrated to highly express NT, and this expression was significantly reduced by cold exposure and norepinephrine. In vivo and ex vivo experiments demonstrated that NT is capable of reducing adaptive thermogenesis in brown adipocytes and favoring the development of an obesogenic phenotype via its specific receptor NTSR2 [64]. As confirmation, treatment with a specific NTSR2 inhibitor peptide enhanced the expression of the pro-thermogenic gene uncoupling protein 1 (UCP-1) and lowered lipid content in the adipose tissue of treated mice [71]. Finally, long-term NTSR2 inhibition in experimentally obese mice reduced adiposity, and improved glucose and insulin metabolism and adipose tissue metabolic function, as evaluated by greater UCP and peroxisome proliferator-activated receptor gamma (PPAR-*γ* expression 71).

Indeed, if on the one hand, NTSR1s are likely involved in promoting inflammatory processes, on the other hand, NTSR2s seem to be important for energy balance regulation in human adipose tissue.

Altogether, data on NT and lipid metabolism show the existence of a complex and integrated network of functions exerted by this small peptide, converging in the modulation of fat homeostasis. NT is involved in the major steps of lipid metabolism in humans as it promotes lipid absorption, emulsification, digestion, and tissue storage.

Moreover, many other processes influencing lipid metabolism are under NT control, such as gastrointestinal mobility, gastro-pancreatic secretion, enterohepatic circulation, microbiota composition, and gut mucosal integrity.

Furthermore, in conditions of chronically elevated caloric load, NT also contributes to adipose tissue inflammatory reactions and metabolic impairment, thus, it not only influences the amount of lipid influx, but also the adipose tissue microenvironment and function.

## 4. NT and Glucose Metabolism

NT is a gastrointestinal neuropeptide and, like other gut hormones, such as glucagon-like peptide 1 (GLP-1) and the gastric inhibitory polypeptide (GIP), its involvement in glucose metabolism and endocrine pancreatic secretion has been extensively investigated. However, available data are multifaceted and not univocal, and the role of pro-NT levels as a predictor of diabetes mellitus in humans is still debated (see also Section 5.2).

Experimental studies show that active sodium-dependent glucose transport in jejunum and ileum promotes NT secretion through a cyclic adenosine monophosphate-dependent subcellular pathway in rats [83]. Conversely, data in humans demonstrates that, unlike what observed for MCFAs and LCFAs, jejunal infusion of glycerol of carbohydrate hydrolysate did not increase NT concentration in healthy individuals [15].

Altogether, the experimental data attribute NT to pro-hyperglycemic effects [84,85,86,87].

High-circulating NT levels increase plasma glucose concentration for mechanisms associated with enhanced glucagon-independent glycogenolysis [84].

In vivo administration of NT in rats caused hyperglycemia [85,86], likely because of presence of a direct glucagonotropic effect of NT, as confirmed in isolated rat islets, where incubation with NT also resulted in increased insulin and somatostatin release [87].

The aforementioned in vitro hyperglycemic effects of NT were glucose-dependent; indeed, while NT stimulated glucagon release in the presence of low glucose concentrations, this effect disappeared in high glucose conditions.

What seemed to be finely regulated in vitro was not confirmed in in vivo studies, and the administration of mimic NT peptides had contrasting results on glucose profile. NT was shown to stimulate insulin secretion at low glucose concentrations but to inhibit glucose-induced insulin release in animal models [88].

Finally, NT actions take to mechanisms associated with insulin sensitivity. In particular, the expression of NTSR3/sortilin was detected in adipocytes and myocytes; in these cells, the hyper-activation of the NT/NTSR3 axis downregulated the translocation of the insulin-sensitive glucose transporter 4 (GLUT4) [89,90], thus providing further molecular basis for the development of insulin resistance associated with elevated pro-NT levels.

## 5. NT in the Pathophysiology of Insulin Resistance-Related Disorders

### 5.1. Obesity

From a general perspective, NT and the NT/NTSR pathway are directly involved in body weight control, mainly (1) by modulating brain circuits responsible for food intake and energy expenditure, and (2) by promoting gut lipid absorption.

Indeed, in the lateral hypothalamic area, as a neurotransmitter of the ghrelin and leptin systems, central NT suppresses hunger and food intake [31,32], whereas peripheral NT, released by gut neuroendocrine cells in response to fatty meal ingestion, contributes to lipid absorption, without affecting food intake [41].

Thus, central and peripheral NT seem to exert opposite effects on body weight control: whether these two systems interact and influence each other is unknown.

As for peripheral NT—the focus of this review—it has been demonstrated that the increase in plasma pro-NT during a fatty meal reflects food lipid content and parallels the rise of blood triglycerides after the meal [49]. As previously mentioned, the NT/NTSR axis promotes body weight gain during HFD regimens in experimental models [34], and most importantly, it drives lipid accumulation in ectopic sites, i.e., in the liver and visceral fat compartments, overall exerting a negative impact on metabolism and cardiovascular risk [41,81,91,92].

Studies show that high fasting pro-NT levels correlate with body weight gain and with the development of dysmetabolic disorders later in life, in both adults [93] and pediatric populations [94]. Overweight and obese children with higher basal pro-NT concentrations displayed a greater trend towards further body weight increase, irrespective of the presence and severity of insulin resistance at baseline. In these subjects, basal pro-NT is associated with the presence of an altered lipid profile, predicted weight gain, and the development of alterations in glucose-insulin metabolism, such as impaired *β*-cell function to compensate for insulin resistance later in life [84].

Unlike what has been reported in the longitudinal setting, data on the relationship between pro-NT and obesity in cross-sectional observations are contrasting [41,91,94].

The effects of peripheral NT on body weight gain are likely driven by the concurrent onset of metabolic alterations in individuals with greater plasma pro-NT levels. In line with this assumption, plasma pro-NT does not correlate cross-sectionally with raw indicators of total body mass per se, such as waist circumference and body mass index (BMI), in either obese [81], overweight, or normal weight individuals [93,94,95].

However, pro-NT is directly associated with the presence of metabolic syndrome, type 2 diabetes, and NAFLD in all of these adult populations [81,91,92]. Similarly, in children, no relationship was found between fasting pro-NT and the presence of obesity in the cross-sectional setting, but those belonging to the highest subgroup of pro-NT concentrations kept accumulating extra weight during their life and developed metabolic impairment [94].

In a newly published study, pharmacological blockade of pro-NT was proven to serve as an anti-obesity therapy [96]. In mice with experimentally induced obesity, treatment with a monoclonal antibody against pro-NT after switching to a chow diet resulted in more accentuated weight loss, and hepatic and adipocyte triglyceride clearance in comparison with a control therapy. Interestingly, antibody-based blockade of pro-NT also resulted in altered behavior, pointing to complex interplay with central mechanisms. The greater weight loss induced by pro-NT inhibition was sustained and remained significant at the long-term follow-up [96].

Moreover, other investigations show that NT administration reduced the expression of thermogenic genes in brown adipose tissue explants, while the blockade of NTSR2 promoted thermogenesis and energy expenditure [97]. These findings provide an additional explanation, besides NT-mediated food lipid absorption, of the relationship between pro-NT levels and the onset of obesity and related metabolic disorders.

It is undoubtable that overall NT effects on energy balance is intricate; moreover, it likely reflects the different functions exerted by NT at the central or peripheral level. Studies on NT and the regulation of food intake, metabolism, and energy expenditure point in a different direction. On the one hand, investigations of central NT suggest an anorexigenic effect of this peptide; on the other hand, its circulating levels and hormone function strongly point toward its involvement in weight gain and fat mass accumulation in the long term.

Importantly, chronic exposure to high NT levels was shown to control the expression of NTSR1 in cultured neurons, exerting negative feedback on brain NTSR1 levels [98].

Whether NT just has divergent effects in the brain and periphery, or whether we are facing a paradigm of *hyper-neurotensinaemia* with chronic central NTSRs stimulation, receptor internalization, and *“neurotensin resistance”*—as in other hormones controlling energy balance, i.e., leptin [99]—which might explain high peripheral pro-NT in metabolic diseases, has still to be elucidated.

### 5.2. Diabetes Mellitus

Several cross-sectional [81,91,92,95] and longitudinal [93,100,101] investigations showed the existence of a positive association between high fasting pro-NT levels and the presence or development of T2DM.

The first demonstration of a relationship between pro-NT and T2DM was obtained in the population-based Malmö Diet and Cancer Study, accounting for over 4600 participants undergoing plasma pro-NT measurements at baseline and follow-ups for over 13 years [93]. In the cross-sectional phase of this investigation, pro-NT positively correlated with fasting blood glucose and insulin; in non-diabetic individuals, fasting pro-NT was associated with incident diabetes later in life. Conversely, in the elderly population enrolled in the Malmö Preventive Project, pro-NT only predicted incident diabetes in the female cohort, although an association was found with the onset of cardiovascular disease (CVD) regardless of sex [101].

In cross-sectional studies, fasting pro-NT was associated with the presence of T2DM, both in non-obese [81] and obese [92] individuals, and correlated with indicators of poor glucose control and insulin resistance, such as higher glycosylated hemoglobin, fasting blood insulin, and HOMA-IR [92].

Recently, results from the “Reasons for Geographic and Racial Differences in Stroke (REGARDS)” [100] study showed that basal pro-NT levels are associated with the development of metabolic syndrome, particularly with low HDL cholesterol and abnormal glucose metabolism, as considered as either high fasting blood glucose or stable treatment with antidiabetic agents, after almost 10 years of follow-up. However, no independent association was found between basal pro-NT and incident diabetes [100].

In this study, the relationship between pro-NT and the onset of metabolic syndrome was mediated by higher HOMA-IR in patients with greater pro-NT at baseline, suggesting that most of the influence exerted by pro-NT on metabolism is likely to be attributable to its negative impact on insulin sensitivity.

Besides T2DM, we recently explored pro-NT in type 1 DM (T1DM) and showed that elevated fasting pro-NT concentrations were associated with poor glycemic control and with the development of visceral adiposity, insulin resistance, and features of metabolic syndrome, also predicting a higher CV risk score at the 10-year follow-up in this population [102].

In conclusion, it is undoubtable that an association exists between greater fasting pro-NT levels and alterations to glucose-insulin metabolism. Evidence from clinical studies suggest that this association is mostly evident in female populations and largely attributable to an underlying condition of insulin resistance.

### 5.3. Non-Alcoholic Fatty Liver Disease (NAFLD) and Liver Cancer

NAFLD is one of the most harmful and underestimated metabolic complications of obesity; it increases the risk of T2DM and diabetes’ complications and represents an independent risk factor for cardiovascular morbidity and mortality [103].

For the strong association between insulin resistance, metabolic diseases, and NAFLD, the acronym MAFLD, Metabolic (dysfunction) Associated Fatty Liver Disease, has been recently proposed to define this condition [104]. In particular, MAFLD is defined as the coexistence of hepatic steatosis and overweight/obesity and/or diabetes mellitus, or evidence of metabolic dysregulation in lean individuals; this definition does not exclude other etiologies of liver disease, i.e., excessive alcohol consumption or viral hepatitis [104].

In the last decade, NAFLD has become the most common chronic liver diseases worldwide, representing a major social and economic cost for governments and populations [105,106]. However, no effective therapy beyond lifestyle intervention has yet been identified.

NT has been shown to enhance hepatic fat accumulation in experimental models of NT gene knockout and overexpression [41]. When fed with HFD, mice lacking whole-body NT genes were protected from obesity-induced hepatosteatosis, as evaluated by liver histology and liquid chromatography–mass spectrometry for cholesterol and triglycerides. In NT−/− mice fed with HFD, fecal triglyceride content was increased by almost 25% [41]. Similar results were obtained in *Drosophila*, where human full-length NT cDNA expression in midgut enteroendocrine cells resulted in increased lipid accumulation in the midgut, and oenocytes, specialized hepatocyte-like cells [41].

A couple of years later, our group produced the first evidence on the existence of a relationship between high fasting plasma pro-NT and the presence/degree of non-alcoholic fatty liver (NAFL) and steatohepatitis (NASH) in humans [102]. Pro-NT directly correlated with greater steatosis, inflammation, and fibrosis grade at the liver biopsy in morbidly obese individuals, independent of potential metabolic confounders. Comparable results were obtained in non-obese individuals with and without T2DM, where greater plasma pro-NT was associated with the presence of NAFLD in a liver ultrasound after adjusting for metabolic covariates [41].

Recently, Villar et al. confirmed the association between greater circulating pro-NT and histologically proven NAFLD in an additional cohort of obese individuals [107].

Furthermore, Dongiovanni and collaborators demonstrated that polymorphisms of the NT (rs1800832) and NTSR1 gene (rs6090453) are associated with hepatic fibrosis, cirrhosis, and hepatocarcinoma in bariatric patients with NAFLD [108]. Moreover, plasma pro-NT was associated with greater BMI, age, the presence of T2DM, lobular inflammation, and fibrosis score during liver histology. Transcriptomic data revealed that hepatic NT levels correlated with the expression of fibrogenic genes, such as transforming growth factor *β* (TGF *β*), alpha smooth muscle actin 2 (ACTA2), and collagen type I alpha 1 chain (COL1A1) in liver fragments [108]. Indeed, a certain role of the NT/NTSR system in the development or—at least—progression of metabolic liver disease is highly suggested.

In this context, NT may represent a therapeutic target for NAFLD. Intriguingly, *metabolitin*, a newly-identified peptide which was recently shown to improve insulin-resistance and NAFLD in mice, acts by linking the intestinal G-protein-coupled receptor GPRC6A, thus inhibiting gut NT expression, and ultimately decreasing lipid absorption in the small intestine [109]. In line with this finding, treatment with monoclonal antibodies blocking pro-NT on top of a dietary intervention resulted in a greater reduction of intrahepatic fat in experimental obese mice, in comparison to diet and control therapy [96].

NT may influence hepatic metabolic processes at different levels. Besides increasing lipid influx from the gut towards the liver [41], NT controls cell proliferation and an inflammatory microenvironment in the liver [108,110,111,112,113].

Higher tissue-specific NT expression levels were found in hepatic cancer, and local NT correlated with the extension of cancer-associated inflammatory responses [98,100,101,102,103]. NT was shown to induce local inflammation and promote tumor invasion in hepatocellular carcinoma by stimulating local interleukin 8 production, which in turn was associated with the activation of the MAPK and NF-κB pathways [111,112]. NT also seems to act as a co-mitogenic factor in the presence of its receptor NTSR1 and epidermal growth factor (EGF) in tumor slice cultures of fibrolamellar hepatocellular carcinoma [110].

Taken together, available data thus far depict an interplay between NT, enhanced gut fat absorption, and the deposition of lipids into the liver. This comes together with an action displayed by NT in triggering inflammatory processes and cell proliferation.

These mechanisms may explain the relationship between pro-NT and the progression of metabolic liver diseases towards more advanced stages, such as the development of severe liver fibrosis and hepatocarcinoma.

## 6. NT and Cardiovascular Disease

The connection between pro-NT and cardiovascular disease is the most extensively investigated association in epidemiological studies on NT and clinical disorders. Plasma pro-NT has been associated with the presence and development of cardiovascular diseases and with increased cardiovascular mortality, independent of traditional risk factors, in both middle-aged [93,114,115] and elderly [101] populations. Indeed, high fasting pro-NT levels predict the onset of hard cardiovascular outcomes, such as myocardial infarction and ischemic stroke, in large population-based investigations [93,115,116]. While pro-NT levels correlated with left ventricular mass and coronary artery calcium content in the Framingham Heart Study (FHS) Offspring cohort [105], what was less defined was the association between pro-NT and coronary artery disease in cross-sectional [117] and longitudinal clinical studies [116].

Whether the relationship between the NT/NTSR axis and the onset of CVD is independent or is mediated by the association between pro-NT and traditional cardiovascular risk factors—i.e., obesity [93,94], diabetes mellitus [93], and NAFLD [92,107]—is still a matter of study.

Unlike what has been observed for major CV outcomes, data from the REGARDS cohort do not show a correlation between plasma pro-NT at baseline and the development of hypertension at the 9.4 years follow-up [118]. The association between pro-NT and impaired vascular function may be not direct, but rather mediated by the interaction between NT and other biological systems regulating blood pressure and the CV system, i.e., the renin–angiotensin–aldosterone system (RAAS) [119], or antidiuretic hormone [120].

Greater pro-NT levels at baseline predict increased all-cause and cardiovascular mortality in both male and female populations. However, as with T2DM, a trend towards a stronger association between pro-NT and CVD was definitively demonstrated in female populations [114].

Most of the evidence from experimental models showed that NT exerts a direct role in modulating immune-inflammatory responses, finally leading to a chronic pro-inflammatory state, accelerated atherosclerosis, and impaired vascular function.

Mice knocked-out for the NTSR3 gene are protected from obesity-induced metabolic disorders [121]. Genetic variations of the NTSR3 gene are associated with vascular inflammation, atherosclerosis, plaque area, and vascular calcification in animal models [122], which confer an increased risk of coronary artery disease and myocardial infarction in humans, independent from the presence of obesity [122].

Overall, available data thus far suggest that NT directly contributes to the development of chronic inflammation, determining plaque instability, and vascular alterations.

However, it is reasonable to hypothesize that the negative impact of NT on the cardiovascular risk profile is also attributable to the close interaction between the NT/NTSR axis and metabolic pathways mediating traditional cardiovascular factors, such as glucose-insulin metabolism, lipid trafficking, and fat distribution.

## 7. NT as a Therapeutic Target and Screening Tool for Dysmetabolic Conditions: Evidence and Future Perspectives

Despite decades of investigations and the knowledge of NT/NTSR pathways in both the central nervous system and the periphery, the overall potential effects of NT and NT-modulating treatments on metabolism are not fully elucidated, when not, in some cases, even conflicting. This could be mostly attributable to the complexity and variety of NT actions on energy expenditure, food intake, and metabolism, as shown in experimental models. A key element to understand the role of NT in the modulation of metabolism is that this small peptide exerts different effects in relation to the tissue and the receptor that it is mostly expressed in.

In relation to the anorexigenic properties that NT displays via the modulation of central leptin and ghrelin pathways, NT has been proposed as an anti-obesity agent. NT injection in specific areas of the central nervous system or in the periphery through NT PEGylated molecules—allowing it to overcome the obstacle of its short serum half-life—reduced appetite and induced weight loss in obese rodents [123,124].

The metabolically favorable effects of NT appear to be mediated by central NTSR1s: indeed, mice knocked-out for this receptor are less responsive to a reduction in appetite after leptin administration [7] and, in general, display increased food intake and body weight gain when fed with HFD [125].

In line with these data, the administration of a long-acting NT peptide, alone or in combination with the GLP1 receptor agonist liraglutide in mice, reduced body weight after 6 days of treatment via mechanisms involving the NTSR1 and melanocortin pathways in the central nervous system [124]. In NT-treated mice, the decrease in food intake was not due to malaise or nausea, as demonstrated by a taste aversion test [124].

However, it is important to underline that chronically elevated plasma NT levels exert negative feedback on NTSR1 expression in the central nervous system, and induce receptor internalization, likely removing the favorable effects of NTSR1 on metabolism [98].

Conversely, in other experimental models, the deletion of the NTSR2 gene protected obese mice from further weight gain, and improved glucose and insulin metabolism, favoring browning processes in the adipocytes [97].

Finally, the administration of a monoclonal antibody binding to the long fragment NT—or pro-NT—and blocking the link with NTSR1 was demonstrated to favor weight loss and to partially restore metabolic homeostasis in mice with obesity [96]. Treated mice were also less stressed and more active, in line with evidence of a major role of NT in modulating activity and behavior [96].

## 8. Conclusions

NT is a small peptide with broad effects on metabolism; such a wide spectrum of central and peripheral activities translates to important implications on the development of metabolic and cardiovascular disorders.

Indeed, NT seems to represent a connecting link between the central nervous system and the intestines, and the effectors of this neuro–endocrine connection are mostly represented by lipids. Metabolic pathways potentially linking chronic high-circulating NT levels with cardio-metabolic diseases in humans are illustrated in Figure 2:

In physiological conditions, NT acts as a central regulator of whole-body energy balance: it reduces food intake and stimulates active behavior and locomotor activity, through its contribution to neurotransmission within dopaminergic brain areas. Indeed, such crosstalk between peripheral NT, which rises after lipid intake, and central NT-mediated circuits, which control energy expenditure, is the key for fine regulation of body weight and metabolism.

Evolutionarily, NT could be seen as a *thrifty peptide:* being able to specifically sense the intake of energy-dense food, NT could maximize lipid uptake from the gut lumen in conditions of increased energy requests and/or food shortage, modulating appetite and energy expenditure accordingly.

However, chronic elevated fat-rich food intake seems to disrupt this optimized energy control: the overstimulation of the NT/NTSR axis is the driver of peripheral and central mechanisms leading to excessive lipid influx, fat expansion, stress and dysfunction, intraparenchymal lipid accumulation, and metabolic impairment. Moreover, chronically high pro-NT levels downregulate brain NT activity, a model of *NT-resistance* with detrimental consequences on the cardiometabolic risk profile (Figure 2).

Which effect, among the effects that NT displays on body weight and fat accumulation or systemic inflammation/insulin-resistance, is the driver of poor cardio-metabolic outcomes in the presence of high pro-NT, is a matter of debate.

Similarly, whether NT itself or the differential interaction with the type of NTSR is the primary effector of metabolic impairment in the course of NT/NTSR hyperactivation has not yet been fully elucidated.

The NT/NTSR axis represents an intriguing object of study, in relation to the possibility of miming/antagonizing this peptide, and a chance to get further insights on processes connecting the central control of appetite and energy intake to the regulation of nutrient absorption and macronutrient flux at the peripheral level.

## Figures and Tables

**Figure 1 ijms-23-02209-f001:**
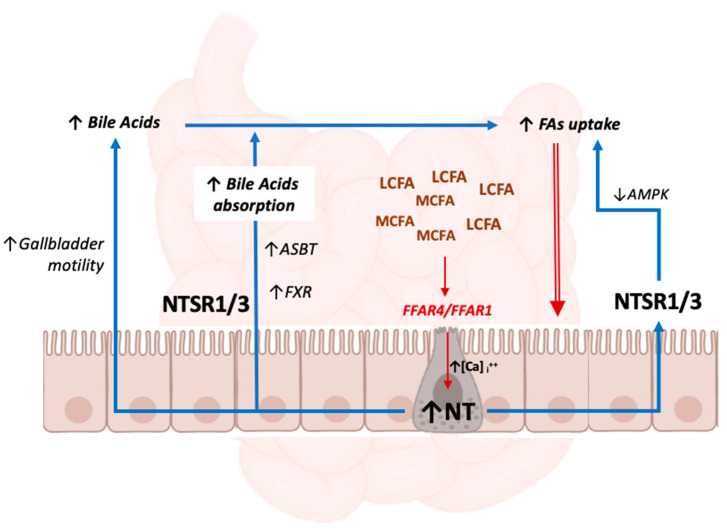
The mechanisms of increased lipid absorption from the gut lumen mediated by intestinal neurotensin. Elevated FA concentrations in the intestinal lumen and the link between LCFAs /MCFAs and FFAR1/FFAR4 lead to increased intracellular Ca^2+^ concentrations in enteroendocrine cells, signaling cascade activation and the secretion of NT. NT directly promotes FA uptake by inhibiting AMPK, and indirectly by favoring bile acids’ absorption though several mechanisms. See Section 3.1, Section 3.2 and Section 3.3 for detailed descriptions and references. Abbreviations: AMPK: 5′ adenosine monophosphate-activated protein kinase; ASBT: apical sodium-dependent bile acid transporter; FA: fatty acid; FFAR1: free fatty acid receptor 1; FFAR4: free fatty acid receptor 4; FXR: farnesoid X receptor; LCFA: long-chain fatty acid; MCFA: medium-chain fatty acid; NT: neurotensin; NTSR1: neurotensin receptor 1; NTSR3: neurotensin receptor 3.

**Figure 2 ijms-23-02209-f002:**
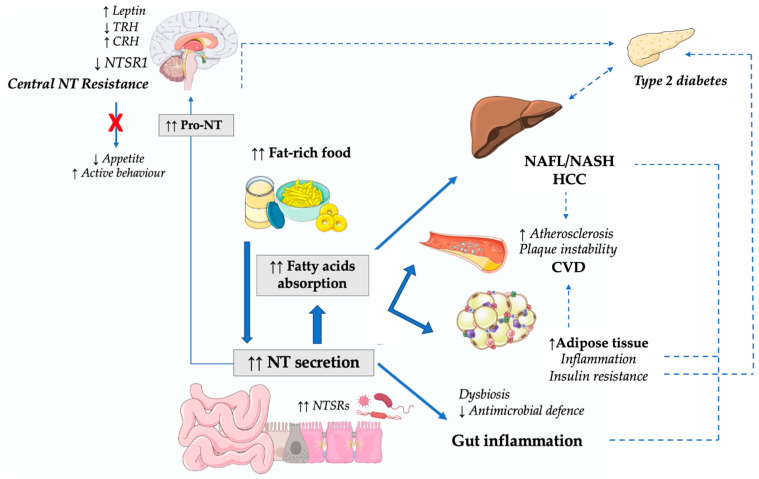
NT in metabolic diseases: potential direct and indirect mechanisms linking high NT to metabolic impairment and cardiovascular disease. Bold arrows indicate direct NT effects, dotted arrows show indirect NT influences on cardio-metabolic risk factors. Abbreviations: CRH: corticotropin-releasing hormone; CVD: cardiovascular disease; HCC: hepato-cellular carcinoma; NAFLD: non-alcoholic fatty liver disease; NASH: non-alcoholic steatohepatitis; NT: neurotensin; NTSR1: neurotensin receptor 1; TRH: thyrotropin-releasing hormone.

## Data Availability

Not Applicable.

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
