# Peer review of "New Insights in the Control of Fat Homeostasis: The Role of Neurotensin"

_ijms, 2022, doi:10.3390/ijms23042209_

Round 1

Reviewer 1 Report

Authors described here roles of neurotensin (NT)  in lipid metabolism citing recent works. This manuscript is an informative and useful review, because many researchers (including me) are interested in effects of endogenous factors such as hormones and cytokines on metabolic regulations. However, many of the paragraphs do not focus on lipid metabolism. For example, the mention of the role of NT in obesity and diabetes is very important, but it is not clear how the effects of NTs on lipid metabolism are involved in the pathogenesis of these diseases. Throughout the paper, a focus on lipid metabolism is needed. In addition, the following points need to be revised.

Majors:

1) Many paragraphs are too small. For example, the first five paragraphs of section 6, "NT and Cardiovascular Disease," are only one sentence long. The paragraph should not be divided into such a number of paragraphs. The paragraphs are so small that they appear to be just a list of research reports. A role of a review article should present the relevance between previous independent research reports. In addition, it is useful for readers to describe briefly what results have been shown in the previous research reports in some paragraphs (ex. the second paragraph in the section 3.4).

2) There are too many ambiguous statements such as "be involved in" and "be associated with" in the lead sentences of paragraphs. Readers understand outline of each paragraph by reading the lead sentence. Thus, authors should mention the direction of NT effects (ex. NT "increases or decreases” and "stimulates or inhibits” something). Due to the ambiguity of the lead sentence, I get the impression that the content of the paragraph itself is ambiguous.

Minors:

1) Line 160-163,Page 4; Is AMPK a negative regulator of NT secretion? In Ref.45, it has been shown that AMPK activation stimulated NT secretion? Please check it.

2) FXR is mentioned, but LXR is not. In cholesterol metabolism, LXR is often mentioned together with FXR. Therefore, if there is no report on LXR, it should be mentioned. In addition, a section on the role of NTs in cholesterol metabolism would be helpful.

Reviewer 2 Report

The review summarized the current understanding of neurotensin in energy balance, particularly fat homeostasis, to increase incidences of metabolic diseases. While a comprehensive and informative review, it needs further organized to meet the publication criteria.   

Reviewer 3 Report

The review article:  New Insight in the Control of Fat Homeostasis: The Role of 2 Neurotensin by Ilaria Barchetta et al. summarizes data on the role of neurotensin and focused on its peripheral activities.

Figure 2, inflammation and insulin, the two “I” may be written in capital letter. CRH seems not to be mentioned in the text of the manuscript. Leptin may be included in the diagram.

“gut lipids absorption“ may be “gut lipid absorption”

“increased lipids concentration“ may be “increased lipid concentrations”

“Intriguingly, the promotion of fat absorption  in relation to food lipid content seems to be at the origin of the strong correlation between high  circulating concentration of pro-NT, the NT stable precursor, and the increased incidence of metabolic disorders, cardiovascular diseases and cancer observed in large population studies.“ this sentence has to be rewritten for clarity.

Which cells produce NT? Is cleavage an extracellular event?

Table 1, is 112 below Hippocampus necessary? Are these mice studies? What do the colours indicate? Central has to be added to the legend.

“The concentration of this short gut peptide, whose structure was identical to the neuro-tensin isolated in bovine hypothalamus, was shown to increase from duodenum to distal 118 ileum [9-10].” Thus, human and bovine NT have the identical amino acid sequence? Is it necessary to refer to bovine?

“Lastly, Fawad et al. demonstrated that the amount of rise of plasma NT after 127 fat ingestion parallels blood triglycerides increase [42], providing the first evidence in hu-128 mans that the increase of NT after lipid ingestion does contribute to the intestinal lipid 129 uptake.” This study only shows an association and does not provide any evidence for the role of NT. Please rewrite.

“Indeed, as an overall result, fat-rich diet ingestion increases pro-NT which, in turn, enhances the activity of gut NTSR1, resulting in augmented fat absorption, visceral fat and body weight “ Can NT and pro-NT activate NTSRs?

“intestinal N cells“ ?

DHA is supposed to be protective regarding metabolic diseases. How does this fit with an induction of NT?

MCFA, PKC and further abbreviations are not explained.

“FFAR1 directly promotes insulin secretion” please explain this in more detail.

Is the NT precursor stored in intracellular vesicles? Where is it processed?

“leading to increased bile  acid uptake via NTSR1 and NTSR3” ?

“Mice knocked-out for the NT gene fed were protected from HFD-induced gut dysbio-216 sis in comparison with the wild type mice,” please correct.

Does NT reduce AMPK in the intestinal cells? When bile acid absorption is induced why do bile acids increase? Is FFAR1/4 also regulated in the epithelial cells?

“such as greater macrophages infiltration and hypoxia,” in adipose tissue? White / brown fat? Subcutaneous / visceral?

“NAFLD/NASH“ either NAFLD alone or NAFL/NASH. See also 5.3. title

“Importantly, chronic exposure to high NT levels” in the periphery?

Please more precisely explain MAFLD

“polymorphisms of the NT (rs1800832) and NTSR1 gene (rs6090453)” do this polymorphisms affect levels of these proteins?

“NT controls cell proliferation and inflammatory micro-environment” In the liver?

“mitogen-activated protein kinase (MAPK) and nuclear factor-kappa B (NF-514 κB) pathways” seems that these abbreviations were defined before.

“metabolic and non-metabolic liver diseases” only metabolic liver disease was described above. Is there a role of NT in other kinds of liver diseases?

Is there a difference between male and female NT levels in the general population?

“without inducing food taste aversion“ this has to be explained in more detail.

Round 2

Reviewer 1 Report

The authors have revised the previous manuscript. However, I strongly recommend to rewrite the entire manuscript. It is not enough only to revise the text in to be accepted.

Reviewer 3 Report

in the definition of MAFLD the word "lean" has to be deleted.